# Descriptive Kernel Convolution Network with Improved Random Walk Kernel

## ABSTRACT

Graph kernels used to be the dominant approach to feature engineering for structured data, which are superseded by modern GNNs as the former lacks learnability. Recently, a suite of Kernel Convolution Networks (KCNs) successfully revitalized graph kernels by introducing learnability, which convolves input with learnable hidden graphs using a certain graph kernel. The random walk kernel (RWK) has been used as the default kernel in many KCNs, gaining increasing attention. In this paper, we first revisit the RWK and its current usage in KCNs, revealing several shortcomings of the existing designs. To this reason, we propose an improved graph kernel $RWK^+$ by introducing color-matching random walks, and derive its efficient computation. We then propose $RWK^+CN$, a KCN that uses $RWK^+$ as the core kernel to learn descriptive graph features with an unsupervised objective, which can not be achieved by GNNs. Further, by unrolling $RWK^+$, we discover its connection with a regular GCN layer, and propose a novel GNN layer $RWK^+Conv$. In the first part of experiments, we demonstrate the descriptive learning ability of our proposed $RWK^+CN$ with the improved random walk kernel $RWK^+$ on unsupervised pattern mining tasks; in the second part, we show the effectiveness of $RWK^+$ for a variety of KCN architectures and supervised graph learning tasks, and demonstrate the expressiveness of our proposed $RWK^+Conv$ layer, especially on the graph-level tasks. Our proposed $RWK^+$ and $RWK^+Conv$ adapt to various real-world applications, including web applications such as bot detection in a web-scale Twitter social network, and community classification in Reddit social interaction networks.

**ACM Reference Format:**

Anonymous Author(s). 2023. Descriptive Kernel Convolution Network with Improved Random Walk Kernel. In *Proceedings of (WWW '24)*. ACM, New York, NY, USA, 12 pages. https://doi.org/10.1145/nnnnnnn.nnnnnnn

## 1 INTRODUCTION

Graph kernels have historically been a popular approach to "flatten" graphs explicitly or implicitly into vector form that many downstream algorithms can more easily handle. While graph kernels exhibit mathematical expressions that lend themselves to theoretical analysis [16], their handcrafted features may not be expressive enough to capture the complexities of various learning tasks on graphs [43]. More recently, graph kernels are superseded by modern GNNs which leverage multi-layer architecture and nonlinear transformations to learn task-adaptive graph representations [60].

Interestingly, GNNs bear a close connection to the Weisfeiler-Leman (WL) graph kernels [48], as well as the related WL graph isomorphism test [53]. In fact, most recent work on the expressive power of GNNs heavily use the $k$-WL hierarchy [28, 47], and others have derived inspiration from it to design novel GNN architectures [5, 33, 36, 58]. The WL kernel, which is quite popular thanks to its attractive linear-time complexity [19], derives its simplicity from iterative neighborhood aggregation, akin to the convolution scheme of message-passing GNNs [17]. This type of connection has been recognized and leveraged in the recent few years to derive a series of "*GNNs meet graph kernels*" style models that bridge these two worlds [2, 6, 8, 11, 13, 27, 29, 31, 40] (see Sec. 2 for detailed related work), named as Kernel Convolutional Networks (KCNs).

RWKs, based on the number of (node label sequences along) walks that two graphs have in common, have been the starting point in the history of graph kernels [16, 23]. Notably, a recent study by Kriege [25] demonstrated that classical random walk kernels with only minor modifications are as expressive as the Weisfeiler-Leman kernels and even surpass their accuracy on real-world classification tasks. Inspired by this connection, our work extends from the random walk neural network (RWNN) of Nikolentzos and Vazirgiannis [40], where each input graph is represented by its RWK similarity to a set of small graphlets (called hidden graphs) that are learned end-to-end by optimizing a classification objective.

In this paper, we deepen the synergy between GNNs and graph kernels, and improve the RWK as utilized within GNNs in a number of fronts. First, toward capturing more representative patterns, we introduce several improvements to the RWK in both effectiveness and efficiency and propose an improved graph kernel $RWK^+$. Second, we propose a *descriptive* kernel convolution network (KCN) $RWK^+CN$ by flipping the objective from a discriminative one to a descriptive one that helps us capture relational patterns in the graph database. What is more, we derive the mathematical connection of $RWK^+$ to layer-wise neural network operators for the first time, which inspires us to propose a novel GNN layer $RWK^+Conv$. Finally, we employ our $RWK^+$ and $RWK^+Conv$ on a suite of real-world tasks for graph data and achieve significant gains. A summary of our contributions is as follows:

- **$RWK^+$ with efficient color-matching:** We identify that the RWK originally developed in RWNN only enforces the same label at the start and the end of two walks while ignoring the intermediates. We reformulate it to count a walk as shared only if all corresponding node pairs exhibit the same node label (i.e. color) at all steps along the walk. While more reflective of graph similarity, RWK with color-matching incurs a memory and computational overhead. Therefore, we propose the improved graph kernel $RWK^+$ through transforming its formulation for efficient computation. In addition, we propose a learnable solution STEPNORM to address the nontrivial task of combining similarity scores across steps with drastically different scales.

- **$RWK^+CN$ learning descriptive hidden graphs**: The original RWNN is trained supervised for graph classification and thus

learns discriminative hidden graphs. We propose RWK$^+$CN with an unsupervised objective, that uses RWK$^+$ as the core kernel and maximizes the total RWK similarity between the input graphs and hidden graphs. The learned hidden graphs are reflective of the frequent walks (i.e. patterns) in the database. To further enhance the descriptive ability, we use additional "structural colors" to help better capture structural similarity between graphs, and enforce a diversity regularization among the hidden graphs to capture non-overlapping subgraphs. Finally, we demonstrate the descriptive learning ability of RWK$^+$CN with our carefully designed testbeds.

• **RWK$^+$Conv, a novel GNN layer**: By unrolling RWK$^+$, we discover that the derivation can be re-written as a sequence (i.e. multiple layers) of graph convolutional operations, connecting with regular GCN layers. By viewing hidden graphs as learnable parameters, we transform the RWK$^+$ algorithm into a novel GNN layer called RWK$^+$Conv. The RWK$^+$Conv layer uses additional element-wise product operation that can potentially bring better expressiveness than the GCN layer.

• **Broad applications of RWK$^+$ and RWK$^+$Conv**: We employ RWK$^+$ as the core kernel inside different KCN architectures and evaluate it on four graph-level tasks: one discriminative (graph classification), and three descriptive (graph pattern mining, graph-level anomaly detection, and substructure counting). It is shown to be improved over the vanilla RWK especially on descriptive tasks. Moreover, we compare our proposed RWK$^+$Conv layer with the GCN layer on node- and graph-level tasks. RWK$^+$Conv outperforms GCN in both tasks, notably by a large margin in graph-level tasks, empirically demonstrating its better expressiveness. It is worth noting that our experiments contain a broad-range of real-world applications, including web applications such as bot detection in a web-scale Twitter social network with a million nodes, and community classification in Reddit social interaction networks.

**Reproducibility:** All source code and datasets are shared anonymously at https://bit.ly/3Fb4ZIf.

## 2 RELATED WORK

**Graph Kernels.** The literature on graph kernels is extensive and well established, thanks to the prevalence of learning problems on graph-structured data and the empirical success of kernel-based methods [26, 39]. A large variety of graph kernels have been developed motivated either by their theoretical properties, or specialization or relevance to certain application domains like biology [22, 42] or chemistry [45]. Those include graph kernels based on shortest paths [4], subtrees [30, 43], graphlets [42, 49], random walks [16, 23], as well as variants such as random walk return probabilities [57], to name a few. A long line of work focused on designing computationally tractable kernels for large graphs with discrete as well as continuous node attributes [9, 15, 35, 49], while those such as the Weisfeiler-Leman (WL) kernel [48] and others [19] gained popularity thanks to linear-time efficiency.

A key challenge with classical graph kernels is lack of learnability; today's graph neural networks (GNNs) are able to learn feature representations that clearly supersede the fixed feature representations used by graph kernels. At the same time, several connections can be drawn between graph kernels and GNNs, such as the similarity between the neighborhood aggregation of the WL kernel

(a.k.a. color refinement) and the scheme of message-passing GNNs [17]. We discuss below recent line of work that tap into the synergy between graph kernels and GNNs to harvest the best of both worlds.

**Synergizing Graph Kernels and GNNs.** While many works bridge graph kernels with GNNs, they have clear distinctions. Coined as Convolutional Kernel Networks [32], and others in similar lines [6, 31], introduce neural network architectures that learn graph representations that lie in the reproducing kernel Hilbert space (RKHS) of graph kernels. Others design new classes of graph kernels using GNNs [11, 18]. In contrast, and closest to our work, coined very similarly as Graph Kernel Convolution Networks (KCNs) [8] and various others [13, 27, 40] integrate a graph kernel *into* GNN architectures. In other words, they show how to realize a given graph kernel with a GNN module, which in effect unlocks end-to-end learnability for the graph kernel. We provide further background on KCNs in Sec. 3.1. Finally, while different in focus, there is also noteworthy work exploiting graph kernels for pre-training GNNs [37], or to extract preliminary features that are passed onto CNNs [38].

## 3 KERNEL CONVOLUTION NETWORKS WITH RANDOM WALK KERNEL AND BEYOND

Kernel Convolution Network (KCN) [8, 13, 40] that convolves the input graph with learnable hidden graphs using a certain graph kernel has gained increasing attention recently, as it offers learnability to graph kernels. Given the simplicity of random walk kernel (RWK) and its differentiability, it has been used as the default graph kernel in many KCNs like RWNN [40] and KerGNN [13]. We first introduce notation and background of KCN, along with designing an unsupervised loss for learning descriptive features (Sec. 3.1). Then we revisit the RWK (Sec. 3.2), and discuss the issues of its current usage in KCNs (Sec. 3.3). Next, we introduce color-matching based RWK, along with its efficient computation that shares connection to GNNs (Sec. 3.4). Finally, we discuss how to increase the descriptive ability of the learned hidden graphs in the unsupervised setting (Sec. 3.5).

**Notation:** Let $G=(V(G), E(G), l_G)$ denote an undirected, node-attributed graph with $n$ nodes in $V(G)$, $e$ edges in $E(G)$, and an attribute or labeling function $l_G : V(G) \rightarrow C$ where $C$ can be $\mathbb{R}^d$ for continuous attributes or $\{c_1, ..., c_d\}$ for distinct discrete labels. Let $A_G$ denote the adjacency matrix, and $A_{G \otimes H} := A_G \otimes A_H$ depict the Kronecker product of the adjacency matrices for graphs $G$ and $H$. Let $X_G := [\mathbf{x}_{v_1}, \ldots, \mathbf{x}_{v_n}]^T \in \mathbb{R}^{n \times d}$ be the matrix representation of the node attributes in $G$.

## 3.1 Kernel Convolution Networks

Graph kernels are designed to measure similarity on a pair of graphs. However, they produce fixed handcrafted features. Lei et al. [27] derived the first neural network that outputs the RWK similarity scores between input graph and hidden learnable path-like graphs. Nikolentzos and Vazirgiannis [40] generalized Lei et al. [27] such that the hidden graphs can have any structure without the path constraints. The designed model is claimed to be interpretable as the learned hidden graphs "summarize" the input graphs. Later, Cosmo et al. [8] and Feng et al. [13] extended RWNN [40] to a multi-layer architecture, in which each layer compares subgraphs around each node of the input with learnable hidden graphs. We refer to these models as Kernel Convolution Networks (KCNs) as

they generalize the Convolutional Neural Network (CNN) from the image domain to the graph domain, with the help of a graph kernel. Each layer of the KCN has a number of learnable hidden graphs.

Formally, let $G$ be the input graph with node $v \in V(G)$; let $\mathbf{h}^t(v) \in \mathbb{R}^{m_t}$ be the representation of node $v$ at the $t$-th layer where $m_t$ is the number of learnable kernels in KCN's $t$-th layer for $t > 0$, and $m_0$ be the dimension of original node attributes with $\mathbf{h}^0(v) = \mathbf{x}_v$. Let $W_1^t, ..., W_{m_t}^t$ denote the series of learnable hidden graphs in the $t$-th layer, and $\mathrm{Sub}_G^t[v]$ be the subgraph around node $v$ on $G$ with attributes $\{\mathbf{h}^t(u)|u \in \mathrm{Sub}_G[v]\}$, we have:

$$\mathbf{h}^{t+1}(v) = [\mathcal{K}(\mathrm{Sub}_G^t[v], W_1^{t+1}), \ldots, \mathcal{K}(\mathrm{Sub}_G^t[v], W_{m_{t+1}}^{t+1})] , \quad (1)$$

where $\mathcal{K}$ is the graph kernel used to compute graph similarity. Multi-layer KCNs stack graph kernel computations with layers, and output node representations at each layer which can be used for any downstream task. They exhibit strong representation ability however the output is not interpretable or descriptive. The single-layer KCN, while less expressive, can output meaningful similarity scores for descriptive unsupervised feature learning, which computes graph-level representation directly by:

$$\mathbf{h}(G) = [\mathcal{K}(G, W_1), \ldots, \mathcal{K}(G, W_m)] . \quad (2)$$

*Learning Descriptive Features.* KCNs were originally proposed for supervised learning, as such, the learned hidden graphs are discriminative for classification tasks. We claim that the KCN model can be paired with an unsupervised loss and used to generate *descriptive* hidden graphs instead, which is not achieved by existing GNNs. Given the output of a single-layer KCN model is the similarity scores to each hidden graph, one can train the KCN by maximizing the total similarity score. Specifically, the unsupervised objective is given as:

$$\max_{W_1, ..., W_m} \sum_{i=1}^{m} \mathcal{K}(G, W_i) . \quad (3)$$

With this new objective, the learnable hidden graphs are to reflect or summarize the common patterns of the graph database. Put differently, similarities are maximized when the learned graphs capture frequent structural patterns that the kernel is designed to capture.

## 3.2 Revisiting the Random Walk Kernel (RWK)

RWK has been used in KCNs as the default kernel. It has been originally proposed to compare two labeled graphs by counting the number of common walks on both graphs [16, 23]. Formally, consider a labeled (discrete attribute) graph $G$ such that $l(v)$ represents the label of node $v \in V(G)$. Let $\mathcal{R}_t(G)$ be the set of all $t$-step random walks on $G$. For a random walk $\mathbf{p} = (v_1, v_2, .., v_t) \in \mathcal{R}_t(G)$, let $l(\mathbf{p}) = (l(v_1), ..., l(v_t))$ denote the labels along the walk. Then the $t$-step RWK $\mathcal{K}_{rw}^t(G, H)$ computes the similarity of $G$ and $H$ by counting the common walks as follows:

$$\mathcal{K}_{rw}^t(G, H) = \sum_{i=1}^{t} \lambda_i \sum_{\mathbf{p} \in \mathcal{R}_i(G)} \sum_{\mathbf{q} \in \mathcal{R}_i(H)} \mathrm{I}(l(\mathbf{p}), l(\mathbf{q})) \quad (4)$$

where $\mathrm{I}(x, y)$ is the Dirac kernel where $\mathrm{I}(x, y) = 1$ if $x = y$, and 0 otherwise; and $\lambda_i \in \mathbb{R}$ denotes the weight of the $i$-th step's score.

DEFINITION 1. *(Direct graph product) Given two labeled graphs $G, H$ with labeling function $l$, their direct product is a new graph $G \times H$ with adjacency matrix $A_{G \times H}$, vertices $V(G \times H) = \{(u, v) \in V(G) \times$*

$V(H) \mid l(u) = l(v)\}$ and edges $E(G \times H) = \{((u_1, v_1), (u_2, v_2)) \in V^2(G \times H) \mid (u_1, u_2) \in E(G)$ and $(v_1, v_2) \in E(H)\}$.

Gärtner et al. [16] have shown that for any length $t$ walk, there is a bijective mapping between $\mathcal{R}_t(G \times H)$ and $\{(\mathbf{p}, \mathbf{q}) \in \mathcal{R}_t(G) \times \mathcal{R}_t(H) \mid l(\mathbf{p}) = l(\mathbf{q})\}$. Therefore, Eqn. (4) can be rewritten as:

$$\mathcal{K}_{rw}^t(G, H) = \sum_{i=1}^{t} \lambda_i (\mathbf{1}^T A_{G \times H}^i \mathbf{1}) \quad (5)$$

where $\mathbf{1}$ denotes the all-ones vector of length $|V(G \times H)|$. Note that $A_{G \times H}$ is *not* $A_{G \otimes H}$, where the latter is the Kronecker product of $A_G$ and $A_H$ without enforcing label-matching along the walk.

## 3.3 Issues of Adapting RWK to KCN

The original RWK is designed for labeled graphs and cannot handle graphs with continuous node attributes KCNs are often used for. To that end, Nikolentzos and Vazirgiannis [40] proposed an extension of the RWK in their RWNN. Let $X_G \in \mathbb{R}^{|V(G)| \times d}$ depict the $d$-dimensional continuous attributes for all nodes. For two graphs $G$ and $H$, let $S = X_H X_G^T \in \mathbb{R}^{|V(H)| \times |V(G)|}$ encode the dot product similarity between the attributes of the vertices from two graphs, where $\mathbf{s} := \mathrm{vec}(S)$ is the 1-d vectorized representation of $S$. The authors of RWNN proposed to compute the revised RWK as:

$$\mathcal{K}_{rw-}^t(G, H) = \mathbf{1}^T (\mathbf{s}\mathbf{s}^T \odot A_{G \otimes H}^t) \mathbf{1} , \quad (6)$$

where $\odot$ denotes the element-wise product. The revised kernel computes walks with length exactly $t$ only. Mathematically, the term $\mathbf{s}\mathbf{s}^T$ applies reweighting to $A_{G \otimes H}^t$ such that the $(i, j)$-th element becomes $\mathbf{s}_i \mathbf{s}_j (A_{G \otimes H}^t)_{ij}$ where $(A_{G \otimes H}^t)_{ij}$ is equal to the number of length-$t$ walks from pair of nodes $i$ to pair of nodes $j$ in the Kronecker product graph $G \otimes H$.

Although the proposed adaptation of RWK can handle continuous node attributes, we identify two critical issues with Eqn. (6) that we outline below and later address in Sec.s 3.4 and 3.5, respectively.

**Issue 1: Color mismatch.** Let $\mathbf{p} = (u_1, ..., u_t)$ be a walk on $G$ and $\mathbf{q} = (v_1, ..., v_t)$ be a walk on $H$. Eqn. (6) only considers reweighting the number of walks from $(u_1, v_1)$ to $(u_t, v_t)$, where $(u_1, v_1)$ is the starting pair and $(u_t, v_t)$ the ending pair, without comparing the intermediary nodes along the walk. In essence, their formulation of the RWK is limited to only partially shared walks.

**Issue 2: Inefficient parameterization.** Notice that $\mathbf{s} = \mathrm{vec}(S) = \mathrm{vec}(X_H X_G^T) = \sum_{i=1}^{d}(X_G^{[i]} \otimes X_H^{[i]})$, where $X_G^{[i]}$ denotes the $i$-th column of $X_G$. Using this equality, we can rewrite Eqn. (6) as:

$$\mathcal{K}_{rw-}^t(G, H) = \mathbf{1}^T (\mathbf{s}\mathbf{s}^T \odot A_{G \otimes H}^t) \mathbf{1} = \mathbf{s}^T A_{G \otimes H}^t \mathbf{s}$$

$$= (\sum_{i=1}^{d}(X_G^{[i]} \otimes X_H^{[i]}))^T (A_G^t \otimes A_H^t)(\sum_{i=1}^{d}(X_G^{[i]} \otimes X_H^{[i]}))$$

$$= \sum_{i=1}^{d} \sum_{j=1}^{d}(X_G^{[i]T} A_G^t X_G^{[j]}) \otimes (X_H^{[i]T} A_H^t X_H^{[j]})$$

$$= \mathbf{1}^T (X_G^T A_G^t X_G) \odot (X_H^T A_H^t X_H) \mathbf{1} \quad (7)$$

If $H$ is a learnable hidden graph with parameters $A_H \in \mathbb{R}^{m \times m}$ and $X_H \in \mathbb{R}^{m \times d}$, the effective parameters are merely $X_H^T A_H^t X_H \in \mathbb{R}^{d \times d}$. That is, dimension $d$ is an important degree of freedom for learnability, which can be small for certain real-world graphs.

## 3.4 Color-Matching Random Walks with Efficient Computation

To address **Issue 1**, we propose an improved random walk kernel $RWK^+$ by deriving an effective formulation. First, notice that the original RWK for labeled graphs can be rewritten using the Kronecker product $A_{G \otimes H}$ and one-hot encoded representation of the node labels. We slightly change the $G \times H$ notation by introducing a set of "empty" nodes $\{(u, v) \in V(G) \times V(H) | l(u) \neq l(v)\}$ to the direct product graph $G \times H$. "Empty" nodes do not connect to any node hence this does not change the graph, rather they enlarge the size of $A_{G \times H}$ to be the same as $A_{G \otimes H}$. Given $X_G$ and $X_H$ as one-hot encoding of labels, the similarity matrix $S = X_H X_G^T$ is a binary valued matrix. Then, the following relation can be easily established:

$$A_{G \times H} = \text{diag}(\mathbf{s}) A_{G \otimes H} \text{diag}(\mathbf{s}) = \mathbf{s}\mathbf{s}^T \odot A_{G \otimes H} \,, \quad (8)$$

where $\text{diag}(\mathbf{s})$ denotes a diagonal matrix with $\mathbf{s}$ being the diagonal. Thanks to Eqn. (8), we can rewrite the original RWK in Eqn. (5) as:

$$\mathcal{K}^t_{rw+}(G, H) = \sum_{i=1}^{t} \lambda_i [\mathbf{1}^T (\mathbf{s}\mathbf{s}^T \odot A_{G \otimes H})^i \mathbf{1}] \,, \quad (9)$$

which is slightly different from Eqn. (6) with $\mathbf{s}\mathbf{s}^T$ moving inside the power iteration. Although the derivation starts from labeled graphs, Eqn. (9) can be directly used for continuous attributed graphs without modification. Notice that this new formulation now takes all intermediary node attributes into consideration when comparing walks as intended.

*3.4.1 Reformulation toward Efficient Computation.* The formulation in Eqn. (9) needs to compute the product graph between $G$ and $H$ which is inefficient in both memory and time. We establish an efficient computation by a property of Kronecker product, $(A \otimes B)\text{vec}(S) = \text{vec}(BSA^T)$ [51], and rewrite the main part of Eqn. (9) step by step as follows:

$$\mathbf{1}^T (\mathbf{s}\mathbf{s}^T \odot A_{G \otimes H})^i \mathbf{1}$$
$$= \mathbf{1}^T (\text{diag}(\mathbf{s}) A_{G \otimes H} \text{diag}(\mathbf{s}))^i \mathbf{1}$$
$$= \mathbf{1}^T \text{diag}(\mathbf{s}^{-1}) (\text{diag}(\mathbf{s}^2) A_{G \otimes H})^i \text{vec}(S)$$
$$= \mathbf{1}^T \text{diag}(\mathbf{s}^{-1}) (\text{diag}(\mathbf{s}^2) A_{G \otimes H})^{i-1} \text{diag}(\mathbf{s}^2) \text{vec}(A_H S A_G^T)$$
$$= \mathbf{1}^T \text{diag}(\mathbf{s}^{-1}) (\text{diag}(\mathbf{s}^2) A_{G \otimes H})^{i-1} \text{vec}(S \odot S \odot (A_H S A_G^T)) \quad (10)$$

The LHS thus can be computed *iteratively* by applying colored operations on the RHS repeatedly, using the procedure outlined in Algo. 1 (where transpose is applied to all variables).

---

**Algorithm 1** Fast Color-Matching RWK    {and $RWK^+$Conv}

---

1: **Input:** $G=(A_G \in \mathbb{R}^{n \times n}, X_G \in \mathbb{R}^{n \times d}); H =(A_H \in \mathbb{R}^{m \times m}, X_H \in \mathbb{R}^{m \times d})$; max step $t$;    { H are parameters in $RWK^+$Conv}
2: **Init:** $Y_0 \leftarrow X_G X_H^T, \quad Y \leftarrow Y_0; \{Y_0 \leftarrow \sigma(X_G X_H^T)$ in $RWK^+$Conv$\}$
3: **for** $i = 1$ **to** $t$ **do**
4:    $Y \leftarrow A_G Y A_H^T$
5:    $Y^{(i)} \leftarrow Y_0 \odot Y$
6:    $Y \leftarrow Y_0 \odot Y^{(i)}$
7: **end for**
8: **Return:** $\sum_{i,j} Y_{i,j}^{(t)}$  or  $\sum_{i,j} (\sum_l \lambda_l \cdot Y^{(l)})_{i,j}$

---

**Complexity Analysis.** Let $G$ be the sparse input graph with $n$ nodes and $e$ edges, and let $H$ be the dense hidden graph with $m$ nodes. Eqn. (9) requires the explicit computation of Kronecker product, requiring runtime complexity $O(em^2)$ and memory complexity $O(em^2)$. In contrast, Eqn. (10) has runtime complexity $O(em + nm^2)$ and memory complexity $O(nm + m^2 + e)$.

*3.4.2 Learnable Similarity Normalization.* In Eqn. (9) we compute the random walk similarity score for each step $i$ iteratively. As each step counts the number of shared walks with length $i$, the scale of the similarity score across different steps can be considerably different, underscoring shorter walks. To combine these scores across different steps, we need to normalize the scores, which is nontrivial. To avoid hand-crafted normalization that needs hyperparameter tuning for different input, we introduce STEPNORM that normalizes the score in a learnable way. STEPNORM combines BatchNorm [21] with the sigmoid function, where BatchNorm first standardizes the score to a Normal distribution and then shifts and rescales the score with learnable parameters. Sigmoid further normalizes the score to the range $[0, 1]$. We place STEPNORM between lines 4 and 5 in Algo. 1 to normalize the score step by step, and set $\lambda_l = 1$ for all $l \in [1, t]$.

## 3.5 Enhancing Descriptive Hidden Graphs

Thanks to the unsupervised objective in Eqn. 3 and our $RWK^+$, KCNs can be used to learn descriptive features. To further enhance the descriptive ability of the hidden graphs learned by KCN, we propose $RWK^+CN$, with two more important solutions:

**S1: Additional "Structural Colors".** As discussed under **Issue 2** in Sec. 3.3, the input feature dimension $d$ (i.e. number of node attributes) is an important degree of freedom for learnability for the original RWK in [40]. For RWK with color-matching, input features also play an important role as the similarity matrix $S = X_H X_G^T$ would be sparser with features that better characterize the nodes. Features that characterize structurally similar nodes also enable stronger feature matching at each step of a pair of walks between two graphs. Therefore, we propose to enrich the original node features with additional structural features in unsupervised learning of descriptive hidden graphs. As randomly initialized GNN can produce reasonable features for evaluating similarity in graph generation [50], we generate additional structural features through a fixed randomly initialized GNN to augment the original features.

**S2: Diversity Regularization.** When learning with more than one hidden graph without any constraint, the optimization may end up learning either the most frequent or otherwise very similar patterns. Therefore, we introduce diversity regularization $\mathcal{R}$ toward learning non-overlapping hidden graphs, defined as follows:

$$\mathcal{R}(W_1, \dots, W_m) = \frac{2}{m(m-1)} \sum_{i=1}^{m-1} \sum_{j=i+1}^{m} \mathcal{K}^t_{rw}(W_i, W_j) \quad (11)$$

The overall unsupervised objective becomes maximizing the input graph to hidden graph similarities, while also minimizing the pairwise RWK similarities among the hidden graphs $\mathcal{R}(W_1, \dots, W_m)$.

## 3.6 Connections with GNNs

Moreover, $RWK^+$ shares connections with Graph Convolutional Networks (GCN) [24]. If we view the hidden graph inside $RWK^+$ as learnable parameters, line 4 of Algo. 1 is given as $Y \leftarrow A_G Y A_W^T$,

which shares the *same* formulation as the graph convolutional operation in GCN, ignoring the activation function. Besides convolution-like computation, RWK$^+$ with learnable hidden graph also has a gated element-wise product as in line 5 of Algo. 1.

To demonstrate the connections with GNNs, we propose a novel GNN layer RWK$^+$Conv, based on Algo. 1 (see the gray part). The major differences between RWK$^+$Conv and a normal GCNConv are: (1) element-wise product operation with $Y_0$ motivated from node color matching; and (2) multi-step within a single convolution layer that shares the same parameter $A_H$ and $X_H$. Additionally, we make following changes to turn it into a neural network layer: (1) adding a sigmoid to $Y_0$ to normalize the scale of similarity between 0 and 1; and (2) parameterizing $A_H$ with a fully-connected layer. With the learnable hidden graphs and the additional element-wise product operation, we expect RWK$^+$Conv to bring better expressiveness than the GCN layer. We empirically demonstrate this point in Sec 5.2, across many applications.

## 4 EXPERIMENTS I: UNSUPERVISED PATTERN MINING

Through a series of experiments, we show that RWK$^+$CN can be used for several unsupervised pattern mining tasks, and that each of our proposed solutions contribute to improved performance and descriptive ability. Pattern mining, which is typically a graph algorithm subject matter, is a very difficult task to achieve via machine learning. Since our major purpose is to demonstrate the advantages of RWK$^+$CN over RWNN, they are evaluated on a controlled testbed with ground truth, wherein we understand the nature of the graphs. The detailed descriptions are given in Appx. A.

### 4.1 Task 1: Simple Subgraph Matching

We design two tasks where the subgraphs are easy to learn. The first task aims to show that RWK$^+$CN handles color-matching of every node pair along walks, while RWNN does not. The second task demonstrates that diversity regularization aids with learning non-overlapping hidden graphs. We report the matching accuracy for each experiment, where it is considered as a correct match when the model learns the desired subgraph pattern(s).

**Task 1-1.** We generate a database of 100 bipartite graphs with heterophily, where nodes on two sides of the graph have different colors/labels (e.g. Fig. 4a). We use one hidden graph, and the task is to learn a bipartite core; "butterfly" (Fig. 4b), or a 3-star with core and peripherals with different colors (Fig. 4c). Two different objectives are used; one is to maximize the total similarities from all steps, and another is to maximize the similarity only from the last step.

Table 1 reports the matching accuracies. Our RWK$^+$CN works well even if the similarity is only from the last step, regardless of the number of steps. Since RWK$^+$CN matches the labels of every node pair in each walk, maximizing the similarity from the last step needs to ensure the correctness of matching from previous steps at the same time. Although RWNN works when the similarity is from all steps, it fails when the similarity is from the last step when the number of steps equals 2. This is because the even-step neighbors in a bipartite heterophily graph have the same color.

However, this task is a special case, where the method only needs to realize that the neighbors should have the other color in the

**Table 1: Task 1-1: Simple subgraph matching in bipartite graphs.** Thanks to color-matching, RWK$^+$CN performs well even when the objective is based only on the last step.

| Method | Objective | # of Steps | Acc. |
|---|---|---|---|
| RWNN | Sum of All Steps | 2 | 26% |
| | | 3 | 100% |
| RWNN | Only Last Step | 2 | 0% |
| | | 3 | 100% |
| RWK$^+$CN | Only Last Step | 2 | 100% |
| | | 3 | 100% |

**Table 2: Task 1-2: Simple subgraph matching in triangle chains.** Diversity regularization helps RWK$^+$CN learn non-overlapping hidden graphs.

| # of Hidden Graphs | Method | Diversity | P1 Acc. | P2 Acc. | Both Acc. |
|---|---|---|---|---|---|
| 2 | RWNN | No | 0% | 0% | 0% |
| | RWK$^+$CN | No | 82% | 24% | 12% |
| | | Yes | 72% | 66% | 44% |
| 3 | RWNN | No | 0% | 0% | 0% |
| | RWK$^+$CN | No | 88% | 44% | 32% |
| | | Yes | 76% | 80% | 62% |
| 4 | RWNN | No | 0% | 0% | 0% |
| | RWK$^+$CN | No | 98% | 68% | 66% |
| | | Yes | 84% | 86% | 74% |

learned pattern. That is to say, RWNN still can not solve complicated cases just by summing up the similarity from all steps. As we will see later in this section, while RWNN always learns rudimentary patterns because of ignoring the intermediate nodes in the walks, RWK$^+$CN learns more sophisticated ones by taking it into account.

In the rest of this section, for fair comparison, we use "Sum of All Steps" as the objective for RWNN, and "Only Last Step" for RWK$^+$CN, which performs well and simplifies the optimization.

**Task 1-2.** To test diversity regularization, we generate a database with 100 node-labeled triangle chains, containing two frequent patterns (e.g. Fig. 5a). Each triangle is either pattern P1 (Fig. 5b) with probability 60% or otherwise P2 (Fig. 5c) with lower frequency. The number of steps is set to 3, which is efficient and sufficient to capture both homophily (1-step) and heterophily (2-step) neighbors.

Table 2 reports the results, where accuracy depicts if *both* P1 and P2 are learned by the hidden graphs. Even without diversity regularization, RWK$^+$CN learns the most frequent pattern P1 with high accuracy. When diversity regularization is applied, accuracies for the second frequent pattern P2 and both patterns increase. The increase is larger when RWK$^+$CN is trained more flexibly with a larger number of hidden graphs to be learned. Notably, RWNN with vanilla RWK fails to learn either of the patterns. As it prefers the more frequent colors, it often learns all the node colors to be the same.

### 4.2 Task 2: GED-Based Evaluation

To further show the advantages of RWK$^+$CN, we design two more tasks each with two different testbeds. For evaluation, these experiments consider a database containing 100 identical graphs, which is used as the ground truth, i.e. only one hidden graph is used in both tasks. As the ground truth is more complex than the ones in Task 1, and it is difficult to learn the exact graph, we use graph edit distance (GED) [46] to measure how close the learned hidden graph is to the ground truth (the lower the better). While GED with

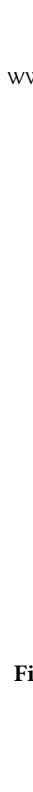
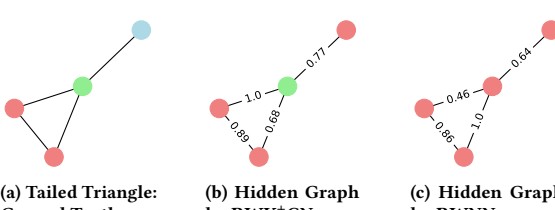

(a) Tailed Triangle: Ground Truth    (b) Hidden Graph by RWK$^+$CN    (c) Hidden Graph by RWNN

| Method | # of Steps | GED w/ Node Labels | $p$-value |
|--------|-----------|--------------------|-----------|
| RWNN | 2 | $3.35 \pm 0.41$ | $3.1e\text{-}05^{***}$ |
| RWNN | 4 | $3.15 \pm 0.34$ | $3.2e\text{-}03^{**}$ |
| RWNN | 6 | $3.25 \pm 0.39$ | $4.7e\text{-}04^{***}$ |
| RWK$^+$CN | 2 | $2.87 \pm 0.58$ | $0.20$ |
| RWK$^+$CN | 4 | $2.82 \pm 0.64$ | $0.33$ |
| RWK$^+$CN | 6 | $\mathbf{2.76 \pm 0.86}$ | - |

(d) Table of results. Lower GED is better.

**Figure 1: Task 2-1: GED-based evaluation on tail-triangles.**

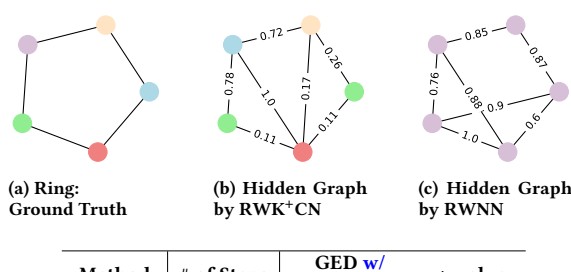

(a) Ring: Ground Truth    (b) Hidden Graph by RWK$^+$CN    (c) Hidden Graph by RWNN

| Method | # of Steps | GED w/ Node Labels | $p$-value |
|--------|-----------|--------------------|-----------|
| RWNN | 2 | $7.08 \pm 0.64$ | $8.4e\text{-}14^{***}$ |
| RWNN | 4 | $7.16 \pm 0.58$ | $2.7e\text{-}14^{***}$ |
| RWNN | 6 | $6.92 \pm 0.75$ | $2.1e\text{-}11^{***}$ |
| RWK$^+$CN | 2 | $5.58 \pm 0.65$ | $0.21$ |
| RWK$^+$CN | 4 | $5.59 \pm 0.73$ | $0.21$ |
| RWK$^+$CN | 6 | $\mathbf{5.46 \pm 0.86}$ | - |

(d) Table of results. Lower GED is better.

**Figure 2: Task 2-1: GED-based evaluation on rings.**

node labels induces a penalty for editing the labels, GED without node labels purely focuses on the graph structure. The first task studies labeled graphs with different number of steps, and shows that RWK$^+$CN outperforms RWNN thanks to color-matching. The second task demonstrates the effectiveness of adding "structural colors", which improves both the learned structure and labels. We report $p$-values based on the paired $t$-test that quantify differences between two GED values statistically.

**Task 2-1.** We design two testbeds using node-labeled tailed triangles and rings, as shown in Fig. 1a and 2a, respectively (best in color). We learn the hidden graph with the same number of nodes as the ground truth graph. Tables 1d and 2d report the GED comparison.

RWK$^+$CN achieves consistently lower GED than RWNN, demonstrating the importance of incorporating color-matching into the RWK. Experiments on both testbeds show that there is no clear choice for the number of steps, i.e., higher is not always better,

**Table 3:** Task 2-2: GED-based evaluation on 3-regular unlabeled graph. Being used as unique identifiers of nodes, structural colors are shown to be as effective as identity matrix.

| Method | Additional Features | GED w/o Node Labels | $p$-value |
|--------|--------------------|--------------------|-----------|
| RWNN | None | $4.38 \pm 0.65$ | - |
| RWNN | Identity | $4.41 \pm 0.56$ | $0.57$ |
| RWK$^+$CN | Identity | $\mathbf{3.89 \pm 0.48}$ | $4.4e\text{-}05^{***}$ |
| RWNN | SC | $4.45 \pm 0.70$ | $0.72$ |
| RWK$^+$CN | SC | $\mathbf{4.10 \pm 0.50}$ | $0.010^*$ |

**Table 4:** Task 2-2: GED-based evaluation on 2-regular labeled graph. Both color-matching and structural colors improve the quality of structure and label learned by hidden graph.

| Method | Additional Features | GED w/o Node Labels | $p$-value w/ Row 1 | $p$-value w/ Row 2 |
|--------|--------------------|--------------------|--------------------|--------------------|
| RWNN | None | $5.25 \pm 0.64$ | - | - |
| RWK$^+$CN | None | $5.02 \pm 0.63$ | $0.049^*$ | - |
| RWK$^+$CN | Identity | $\mathbf{4.81 \pm 0.70}$ | $1.0e\text{-}03^{**}$ | $0.043^*$ |
| RWK$^+$CN | SC | $4.82 \pm 0.70$ | $1.1e\text{-}03^{**}$ | $0.043^*$ |
| **Method** | **Additional Features** | **GED w/ Node Labels** | **$p$-value w/ Row 1** | **$p$-value w/ Row 2** |
| RWNN | None | $7.25 \pm 0.64$ | - | - |
| RWK$^+$CN | None | $6.60 \pm 0.93$ | $1.8e\text{-}04^{***}$ | - |
| RWK$^+$CN | Identity | $\mathbf{6.12 \pm 1.01}$ | $2.9e\text{-}08^{***}$ | $1.8e\text{-}03^{**}$ |
| RWK$^+$CN | SC | $6.25 \pm 1.01$ | $7.2e\text{-}07^{***}$ | $0.015^*$ |

where the p-values are high within RWK$^+$CN. We visualize the learned hidden graphs by removing the edges with the smallest edge weights. Fig. 1b shows that RWK$^+$CN successfully assigns the green node with degree 3 in the correct position. Since the blue node only has degree 1, RWK$^+$CN reasonably learns to maximize the objective by adding one more red node in the hidden graph, which is the most frequent color; in Fig. 1c, RWNN fails to handle the intermediate nodes, and hence includes only the most frequent color in the learned hidden graph. We observe a similar behavior in Fig. 2b and 2c. While RWK$^+$CN pays much attention to learning the correct node labels, RWNN gives a rudimentary result, where all the nodes have the same labels.

**Task 2-2.** Two more testbeds are designed to evaluate structural colors. The number of steps is set to 3, which is effective and efficient. The first database contains 3-regular unlabeled graphs (Fig. 6a). We evaluate RWK$^+$CN and RWNN by GED *without* node labels, focusing on the quality of the learned structure. Our assumption is, if the structural identifiers (node labels) are more unique, then the hidden graph can learn better graph structure. Therefore, we assume identity matrix as the best features in the evaluation, though it is not generalizable to the real datasets. We create the structural colors by a fixed and randomized Graph Attention Networks (GAT) [52]. The results are reported in Table 3. Our proposed RWK$^+$CN using identity matrix as features receives the lowest GED without node labels, as expected. RWK$^+$CN using structural colors has competitive GED compared with using identity matrix, while being more generalizable. These results also empirically prove our assumption that using more unique identifiers as node features helps the hidden graph to learn better structure. In contrast, RWNN fails to utilize the features even if they are extremely informative.

**Table 5: Graph anomaly detection on 10 real-world datasets. Recall is reported. iGAD using our proposed RWK$^+$ as structural feature extractor outperforms original iGAD on all datasets.**

| Dataset | MCF-7 | MOLT-4 | PC-3 | SW-620 | NCI-H23 | OVCAR-8 | P388 | SF-295 | SN12C | UACC-257 |
|---------|-------|--------|------|--------|---------|---------|------|--------|-------|----------|
| iGAD + RWK | 75.1±1.1 | 74.1±0.8 | 77.9±1.2 | 78.6±0.9 | 78.7±1.3 | 78.8±0.3 | 83.1±1.7 | 78.3±1.1 | 79.4±0.6 | 78.0±1.0 |
| iGAD + RWK$^+$ | **76.4±0.6** | **74.3±1.0** | **78.8±1.1** | **79.2±0.5** | **79.5±2.2** | **79.2±0.9** | **84.0±1.4** | **78.5±0.9** | **79.5±1.6** | **79.5±0.7** |

**Table 6: Graph classification on 10 real-world datasets. Accuracy is reported. Although RWK$^+$ is built to capture descriptive features, it is competitive on most datasets.**

| Dataset | MUTAG | D&D | NCI1 | PROTEINS | MUTAGEN | TOX21 | ENZYMES | IMBD-B | IMDB-M | REDDIT |
|---------|-------|-----|------|----------|---------|-------|---------|--------|--------|--------|
| KerGNN + RWK | 81.9±5.3 | **75.2±1.5** | 71.6±2.6 | 75.3±1.2 | 74.4±2.4 | 89.1±0.3 | **47.3±3.9** | 71.2±2.1 | 48.1±2.9 | 77.2±0.5 |
| KerGNN + RWK$^+$ | **83.0±6.4** | 74.8±2.4 | **72.3±1.3** | **76.2±1.2** | **75.1±1.0** | **89.2±0.3** | 44.0±2.7 | **71.6±1.0** | **49.2±0.6** | **77.5±0.6** |

In the second testbed, we study a database containing 2-regular node-labeled graphs (i.e. 6-ring, Fig. 6b), and report the results in Table 4. Only by replacing RWNN with our proposed RWK$^+$CN, the quality of learned hidden graph improves not only on labels, but also on the structure. In addition to the node labels, we further incorporate structural colors into RWK$^+$CN, and find the learned hidden graph improves even better, demonstrating the effectiveness of structural colors. Notably, using identity matrix as features results in only slightly lower GED than using structural colors.

## 5 EXPERIMENTS II: ADAPTING TO VARIOUS APPLICATIONS

In this section, the experiments is composed of two parts. In the first part, we demonstrate that different KCN architectures can perform better by employing our proposed RWK$^+$. In the second part, we compare our proposed RWK$^+$Conv with GCNConv, and empirically show that it has better expressiveness. The details of dataset statistics and hyperparameter search are in Appx. B.

### 5.1 RWK$^+$: Employed to Different Architectures

We conduct three graph learning tasks for evaluating RWK$^+$. Since we focus on improving RWK across many tasks, rather than outperforming task-specific state-of-the-art methods, the experiments concentrate on comparing models using RWK versus RWK$^+$.

#### 5.1.1 *iGAD on Graph Anomaly Detection*.

**Datasets.** We evaluate RWK$^+$ on supervised graph anomaly detection with 10 real-world datasets from PubChem [54], as in [56]. Each graph is a chemical compound and labeled by its outcome from anti-cancer screen tests (active or inactive). The classes are highly imbalanced, where the ratio of the active samples is at most 12%, which are treated as the anomalous cases. We perform 5-fold cross-validation and split 10% of training set as the validation set.

**Settings.** iGAD [56] incorporates RWK as a structural feature extractor to identify graph-level anomalies. For comparison we replace it with RWK$^+$ using STEPNORM, with one-hot node degrees as the node features. Recall is used for both evaluation and model selection, as in the iGAD paper.

**Results.** We report the average performance and standard deviation (stdev) in Table 5. iGAD with our proposed RWK$^+$ outperforms the original model on *all* datasets (*p*-val <0.001). This suggests that the hidden graphs learned through RWK$^+$ are consistently better than

the ones extracted by RWK, assisting iGAD in better pointing out the anomalous graphs that deviate from these patterns.

#### 5.1.2 *KerGNN on Substructure Counting*.

**Datasets.** We evaluate RWK$^+$ on substructure counting with a simulated dataset from Chen et al. [7], following the same setting in [59], and the task is to predict the normalized count of substructures. This dataset includes four tasks, and the evaluation for different tasks are run separately. The dataset provides the training, validation, and testing sets with $1,500/1,000/2,500$ graphs, respectively. One-hot node degrees are used as the node features.

**Settings.** KerGNN [13] uses RWK to compare the similarity between the learnable hidden graphs and the egonets of nodes in a graph. The similarity from different learnable hidden graphs are used as the features for message passing. In this experiment, we replace RWK inside KerGNN with RWK$^+$. Mean absolute error (MAE) is used to measure the accuracy of counts.

**Results.** As shown in Table 7, KerGNN with RWK$^+$ outperforms KerGNN with RWK in 3 out of 4 tasks. STEPNORM is shown to effectively improve the performance of both methods by normalizing the similarity in each step. Without it, the similarity explodes after a number of steps, and is always dominated by the latest step. Although RWK performs better in counting stars, there are few paths to walk within a star, which decreases the necessity of adopting a kernel that is more accurate on similarity.

#### 5.1.3 *KerGNN on Graph Classification*.

**Datasets.** We evaluate RWK$^+$ on graph classification with 10 real-world datasets from TUDataset [34], as in [13]. We use the node labels given by bio-informatics datasets (first 7), and the one-hot node degrees for the social interaction datasets (last 3). We perform 5-fold cross-validation and split 10% of training set as the validation set.

**Settings.** Similar to substructure counting, KerGNN is used with RWK versus RWK$^+$. For fair comparison, STEPNORM is employed for both models. We report average accuracy and stdev.

**Table 7: Substructure counting on a simulated dataset. MAE is reported. RWK$^+$ wins in 3 out of 4 tasks, and STEPNORM is shown to be effective.**

| Task | Triangle | Tailed Tri. | Star | 4-Cycle |
|------|----------|-------------|------|---------|
| KerGNN + RWK | 0.1170 | 0.1346 | 0.1333 | 0.2153 |
| KerGNN + RWK + STEPNORM | 0.1065 | 0.1251 | **0.0999** | 0.2140 |
| KerGNN + RWK$^+$ | 0.1206 | 0.1246 | 0.1750 | 0.2078 |
| KerGNN + RWK$^+$ + STEPNORM | **0.0802** | **0.1240** | 0.1312 | **0.1884** |

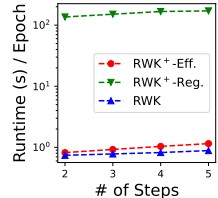

Figure 3: Runtime of KerGNN+RWK$^+$ computed by regular Eqn. (9) vs. efficient Eqn. (10), also compared to vanilla RWK.

**Table 8:** Node classification on 6 real-world datasets. RWK$^+$Conv wins in most tasks on accuracy.

| Dataset | Cora | CiteSeer | PubMed | Chameleon | Squirrel | Actor |
|---------|------|----------|--------|-----------|----------|-------|
| GCNConv | 85.8±0.7 | 73.4±0.5 | **88.0±0.2** | **69.7±0.9** | **55.7±0.4** | 28.3±0.6 |
| RWK$^+$Conv | **88.3±0.5** | **76.7±0.2** | 88.1±0.2 | 69.3±1.9 | 49.7±1.3 | **36.0±0.2** |

**Results.** Table 6 shows that although RWK$^+$ is designed with descriptive, structural graph features in mind, it offers competitive performance on most classification tasks ($p$-val <0.1).

**Scalability.** Finally, we verify RWK$^+$'s scalability empirically, varying (1) the number of hidden graphs and (2) the number of steps on the NCI1 dataset, and report the runtime per epoch during training. In (1), the number of step is set to be 2, and in (2), the number of hidden graphs is set to be 8. As shown in Fig. 3, KerGNN with RWK$^+$ is slightly slower than with RWK, although the overhead is negligible (< 1 sec.), and scales *linearly*, with significant performance gains over regular color-matching computation.

## 5.2 RWK$^+$Conv: Connections with GNNs

To show that RWK$^+$Conv is more expressive than GCNConv, we compare them across both node- and graph-level tasks. In all experiments, we rigorously ensure that both kinds of layers share exactly the same message-passing backbone.

### 5.2.1 Node Classification.

**Datasets.** We evaluate RWK$^+$Conv on node classification with 6 datasets, including homophily graphs (first 3) [55] and heterophily graphs (last 3) [41, 44]. Each dataset is split into 60%/20%/20% for training, validation, and testing, respectively.

**Results.** In Table 8, although expressiveness is not the key in the node-level task, RWK$^+$Conv still has competitive or better performance than GCNConv in most datasets. We also report the run time per epoch on the largest dataset PubMed in Table 9, where RWK$^+$Conv only creates negligible computational overhead.

### 5.2.2 Twitter Bot Detection.

**Datasets.** We evaluate RWK$^+$Conv on a web application, namely bot detection in the TwiBot-22 dataset [14]. This dataset contains a web-scale Twitter social network with one million users, where 86% of them are human, and the rest 14% are bots. We keep only the edges with types "followed" and "following", and make it undirected. The node features are embeddings of user descriptions, transformed by BERT [10]. The dataset provides the training, validation, and testing sets with 70%/20%/10% nodes, respectively.

**Results.** In Table 10, we find that RWK$^+$Conv outperforms GCNConv on F1-score. This suggests that RWK$^+$Conv has better ability to detect the bots by better utilizing the graph structure.

**Table 9:** Runtime of RWK$^+$Conv, with negligible overhead.

| Step Length | 2 | 3 | 4 | 5 |
|-------------|---|---|---|---|
| GCNConv | 0.0269 | - | - | - |
| RWK$^+$Conv | 0.0371 | 0.0464 | 0.0522 | 0.0619 |

**Table 10:** Twitter bot detection on a real-world web-scale dataset. RWK$^+$Conv wins on F1-score.

| Dataset | TwiBot-22 |
|---------|-----------|
| GCNConv | 53.7±0.2 |
| RWK$^+$Conv | **55.0±0.2** |

**Table 11:** Graph regression and classification on 3 real-world datasets. RWK$^+$Conv wins in all tasks.

| Dataset | ZINC | ogbg-molhiv | ogbg-pcba |
|---------|------|-------------|-----------|
| Metric | MAE ↓ | ROC-AUC ↑ | AP ↑ |
| GCNConv | 0.3258±0.0067 | 76.06±0.97 | 20.20±0.24 |
| RWK$^+$Conv | **0.2082±0.0025** | **78.61±0.61** | **24.90±0.12** |

### 5.2.3 Graph Regression and Classification.

**Datasets.** We evaluate RWK$^+$Conv on graph regression and classification with three real-world datasets, namely ZINC [12], ogbg-molhiv and ogbg-pcba [20]. Notice that we do not use edge features.

**Results.** In Table 11, we find that RWK$^+$Conv outperforms GCNConv significantly in all datasets and tasks. This empirically demonstrates the better expressiveness of RWK$^+$Conv than GCNConv.

### 5.2.4 Summary and Future Work.

All results strongly suggest the better expressiveness of our proposed RWK$^+$Conv, especially on graph-level tasks, and its connection to GCN motivates novel convolutional layers for better model design. This offers a direction with large potential to investigate further in the future. We also want to point out that the current design of the RWK$^+$Conv does not take edge features into consideration. Extending it to handle edge features by matching edge colors at every step of random walk could be a potential future work.

## 6 CONCLUSION

In this paper, we first presented RWK$^+$, an improved random walk kernel with end-to-end learnable hidden graphs that can be used by various KCNs. RWK$^+$ incorporates color-matching along the walks that we showed can be efficiently computed in iterations, and combines similarities across steps in a learnable fashion. We then proposed RWK$^+$CN, a KCN that learns descriptive hidden graphs with an unsupervised objective and RWK$^+$. Thanks to additional "structural colors" and diversity regularization, it learns hidden graphs that better reflect the frequent and distinct graph patterns. Moreover, based on the mathematical connection of RWK$^+$ with GNNs, we propose a novel GNN layer RWK$^+$Conv, that extracts expressive graph representations. Experiments showed RWK$^+$'s descriptive learning ability on various unsupervised graph pattern mining tasks, as well as its advantages when employed within various KCN architectures on several supervised graph learning tasks. Furthermore, we showed that our proposed RWK$^+$Conv layer outperforms GCN, especially in the graph-level tasks by a large margin.

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

# A UNSUPERVISED PATTERN MINING

As the optimization is prone to local minima, we randomly initialize the parameters several times and report the average result. The random initialization is done on the parameters of hidden graph(s) and its/their node features. The parameters of hidden graphs are uniformly initialized between −1 to 1. The parameters of the node features of the hidden graphs are uniformly initialized between 0 to 1. Moreover, we have two differences from RWNN. First, RWNN uses the ReLU function right after constructing the hidden graphs, which makes the gradients of edges, whose weights are initialized with negative values, become zero. We thus replace the ReLU function with Sigmoid function to properly learn the hidden graphs. Second, in order to prevent the model stuck at local minimums, we use SGD with a large momentum as the optimizer.

## A.1 Task 1: Simple Subgraph Matching

Figure 4 shows an example of generated graphs in the database and the ground truth graphs in Task 1. The number of nodes on each side is randomly chosen from $[5, 7]$. They are all complete bipartite graphs. The parameters are randomly initialized for 50 times.

Figure 5 shows an example of generated graphs in the database and the ground truth graphs in Task 2. The number of triangles for each chain is randomly chosen from $[3, 5]$. The color set of each triangle is either red-red-blue (P1, Figure 5b), or purple-purple-green (P2, Figure 5c). The parameters are randomly initialized for 50 times.

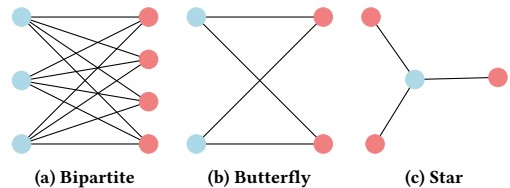

**(a) Bipartite**          **(b) Butterfly**          **(c) Star**

**Figure 4: Simple Subgraph Matching in Bipartite Graphs**

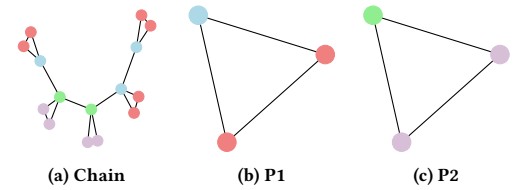

**(a) Chain**          **(b) P1**          **(c) P2**

**Figure 5: Simple Subgraph Matching in Triangle Chain**

## A.2 Task 2: GED-Based Evaluation

We use GED with edge weights, and normalize the edge weights of learned hidden graphs into $[0, 1]$ before computing GED. Given an edge with weight $w$, the cost of removing it is $w$, and the cost of fulfilling it is $1 - w$. The cost of changing the node label is 1. In our experiments, we do not need to add or delete node(s).

A sparsity (L1) loss of the hidden graphs is used and tuned to prevent learning trivial solutions. In both Task 2-1 and Task 2-2,

the parameters are randomly initialized for 50 times. For each of the tasks, we use the same initialization set to run the experiment for several times. For Task 2-2, the sparsity loss is also adopted on both hidden features, to avoid the local minimums. Figure 6 shows the ground truth graphs of Task 2 in GED-based evaluation.

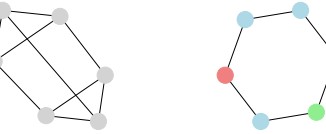

**(a) 3-Regular Graph w/o Colors**          **(b) 2-Regular Graph w/ Colors**

**Figure 6: Ground Truth Graphs in GED-Based Evaluation**

# B REPRODUCIBILITY

## B.1 Configurations

The experiments are conducted on a stock Linux server with an NVIDIA RTX A6000 GPU.

## B.2 Dataset Statistics

We report the dataset statistics of each application in Table 14, 15, and 16. The "REDDIT" dataset, which originally includes more than 200K graphs, is down-sampled for fast evaluation with preserved class ratio.

**Table 12: Dataset statistics of node classification.**

| Dataset | Cora | CiteSeer | PubMed | Chameleon | Squirrel | Actor |
|---|---|---|---|---|---|---|
| # of Nodes | 2, 709 | 3, 327 | 19, 717 | 2, 277 | 5, 201 | 29, 926 |
| # of Edges | 5, 429 | 4, 732 | 44, 338 | 36, 101 | 216, 933 | 7, 600 |
| # of Classes | 7 | 6 | 3 | 5 | 5 | 5 |

## B.3 Search Space of Hyperparameters

**Table 13: Search space of hyperparameters.**

| Application | Hyperparameter Configurations |
|---|---|
| Anomaly Detection | Epoch = 200, lr = 0.001, # of Subgraphs = [8, 16], Subgraph Size = [5, 10], # of Steps = [2, 3] |
| Substructure Counting | Epoch = 500, lr = 0.01, # of Subgraphs = 8, Subgraph Size = 6, Egonet Size = 6, hop = 1, # of Steps = [2, 3], # of Layers = 1 |
| Graph Classification | Epoch = 200, lr = 0.01, # of Subgraphs = 8, Subgraph Size = 6, Egonet Size = 10, hop = 1, # of Steps = [2, 3], # of Layers = 1 |

In Table 13, We report the search space we use for hyperparameter search in each application in Sec. 5.1. Most hyperparameters follow the default settings in iGAD [56] and KerGNN [13], since they are proposed to improve the overall performance. For substructure counting, in order to show that better extracted features can be used more easily to solve the task, we limit the model to be simple, as it is commonly done in linear probing [1, 3]. Since the largest degree in the substructure counting dataset is 6, we set the

learnable subgraph size to 6 as well. For graph classification, we use the default subgraph and egonet sizes from the original paper.

In Sec. 5.2, we run all the experiments with three random seeds and report the average. In node classification, for homophily graphs, the random walk step length is set to 4; for heterophily graphs, TwiBot-22 and graph-level tasks, it is set to 2. For node classification and the ogbg-molhiv dataset, the number of layers is set to 2; for the ZINC and ogbg-pcba datasets, it is set to 6.

**Table 14: Dataset statistics of graph anomaly detection.**

| Dataset | MCF-7 | MOLT-4 | PC-3 | SW-620 | NCI-H23 | OVCAR-8 | P388 | SF-295 | SN12C | UACC-257 |
|---|---|---|---|---|---|---|---|---|---|---|
| # of Graphs | 27,770 | 39,765 | 27,509 | 40,532 | 40,353 | 40,516 | 41,472 | 40,271 | 40,004 | 39,988 |
| # of Anomalies | 2,294 | 3,140 | 1,568 | 2,410 | 2,057 | 2,079 | 2,298 | 2,025 | 1,955 | 1,643 |
| Avg. # of Nodes | 26.4 | 26.1 | 26.4 | 26.1 | 26.1 | 26.1 | 22.1 | 26.1 | 26.1 | 26.1 |
| Avg. # of Edges | 28.5 | 28.1 | 28.5 | 28.1 | 28.1 | 28.1 | 23.6 | 28.1 | 28.1 | 28.1 |

**Table 15: Dataset statistics of substructure counting.**

| Dataset | Task Semantic | # of Tasks | # of Graphs | Avg. # of Nodes | Avg. # of Edges |
|---|---|---|---|---|---|
| CountingSub. | Normalized number of substructures | 4 | 1,500/1,000/2,500 | 18.8 | 62.6 |

**Table 16: Dataset statistics of graph classification.**

| Dataset | MUTAG | D&D | NCI1 | PROTEINS | MUTAGEN | TOX21 | ENZYMES | IMBD-B | IMDB-M | REDDIT |
|---|---|---|---|---|---|---|---|---|---|---|
| # of Graphs | 188 | 1,178 | 4,110 | 1,113 | 4,337 | 8,169 | 600 | 1,000 | 1,500 | 10,155 |
| # of Classes | 2 | 2 | 2 | 2 | 2 | 2 | 6 | 2 | 3 | 2 |
| Avg. # of Nodes | 17.9 | 284.3 | 29.9 | 39.1 | 30.3 | 18.1 | 32.6 | 19.8 | 13.0 | 23.9 |
| Avg. # of Edges | 19.8 | 715.7 | 32.3 | 72.8 | 30.8 | 18.5 | 62.1 | 96.5 | 65.9 | 25.0 |

