# OpenReview forum: "Descriptive Kernel Convolution Network with Improved Random Walk Kernel"
_ACM.org/TheWebConf/2024/Conference — TheWebConf24_

### Official Review · Reviewer_a4ji · 2023-11-21

**Novelty:** 4
**Technical Quality:** 4

**Review:**

**Summary:**
This paper introduces an improved graph kernel called RWK$^+$  with color-matching random walks. RWK$^+$ is used in RWKCN, a Kernel Convolution Network (KCN) for unsupervised graph feature learning. The paper also connects RWK$^+$  to a GGN layer, proposing RWK$^+$ Conv. Experiments demonstrate RWK$^+$ CN's descriptive learning ability and RWK$^+$ 's effectiveness in various tasks, especially for graph-level tasks. RWK$^+$  and RWK$^+$ Conv are adaptable to real-world applications like Twitter bot detection and Reddit community classification.

**Strength**
1. The observations and proposed ideas regarding RWK appear to be sound.
2. The introduction of Color-Matching Random Walks is novel and intriguing, and experimental results confirm their effectiveness.
3. The paper is well-written and easy to read.


**Weakness**
1. The complexity of Algorithm 1 is quadratic with respect to $m$, which seems challenging to run on large-scale graphs, such as those with millions of edges.
2. RWK$^+$Conv is compared only to GCN, and there are some more advanced GNNs, such as GCNII.

I am not familiar with this direction and will discuss it with other reviewers.

**Questions:**

Please refer to the Weaknesses as described above for details.

**Reviewer Confidence:**

2: The reviewer is willing to defend the evaluation, but it is likely that the reviewer did not understand parts of the paper

**Scope:**

4: The work is relevant to the Web and to the track, and is of broad interest to the community

---

### Official Review · Reviewer_DGQD · 2023-11-22

**Novelty:** 6
**Technical Quality:** 5

**Review:**

This paper focuses on the application of graph kernels and specifically Random Walk Kernel (RWK) in kernel convolution networks (KCNs). The authors propose an improved graph kernel RWK+, by introducing color-matching random walks, and derive its efficient computation. They also designed a KCN that uses RWK+ as the core kernel to learn descriptive graph features with an unsupervised objective. Further, by unrolling RWK+, they propose a new GNN layer RWK+Conv.

The paper is well-structured and written. The problem is interesting and in demand. Also, various experiments are conducted, which are discussed in detail. The resources for reproducing the results are also provided. There are some issues with the presentation of the paper, which are discussed in the next part.

**Questions:**

There are some suggestions to improve the presentation of the content:
1.	Page 3, line 240, lacks a more detailed mathematical definition of “learnable hidden graphs.”
2.	Also, in line 241, what is “the subgraph around node G”, please give a math definition.
3.	There’s a typo in line 66 (and more in the paper):  It should be “Weisfeiler-Lehman”
4.	Figure 4 is discussed before figures 1,2, and 3. It would be helpful to mention the figures in the text in ascending order. Also, it is better to move it to the main content rather than the appendix.
5.	Make sure to address different sections of the Appendix in the main content.
6.	Make the font consistent for all the text, notations, figures, tables, captions, etc. Right now, there is a lot of discrepancy.
7.	It is good to move the description of the datasets to the main content.
8.	Tables 14, 15, and 16 are too small and difficult to read.

**Reviewer Confidence:**

3: The reviewer is confident but not certain that the evaluation is correct

**Scope:**

4: The work is relevant to the Web and to the track, and is of broad interest to the community

---

### Official Review · Reviewer_n9Xr · 2023-11-24

**Novelty:** 3
**Technical Quality:** 4

**Review:**

This paper introduces an improved graph kernel called RWK+ and a Kernel Convolution Network (KCN) that uses it to learn descriptive graph features with an unsupervised objective. The proposed RWK+ kernel introduces color-matching random walks and combines similarities across steps in a learnable fashion. RWK+CN uses RWK+ as the core kernel to learn descriptive hidden graphs with an unsupervised objective and diversity regularization. The experiments conducted in the paper show the descriptive learning ability of RWK+ on various unsupervised graph pattern mining tasks and its advantages when employed within various KCN architectures on several supervised graph learning tasks. Furthermore, the proposed RWK+Conv layer outperforms GCN, especially in the graph-level tasks by a large margin.

**Questions:**

Strengths:
S1. The proposed RWK+ kernel introduces color-matching random walks and combines similarities across steps in a learnable fashion, which improves the descriptive learning ability of the kernel.

S2. The proposed RWK+CN uses RWK+ as the core kernel to learn descriptive hidden graphs with an unsupervised objective and diversity regularization, which enables the method to learn more informative graph representations.

S3. The paper provides a detailed analysis of the proposed method on various datasets. The proposed method is also evaluated on various unsupervised graph pattern mining tasks and several supervised graph learning tasks, and it achieves state-of-the-art or competitive performance on most of them.

S4. The paper explores the mathematical connection between RWK+ and GNNs, which provides a theoretical foundation for the proposed method.

Weakness:
W1: What is GCNConv method? Is it the GCN method mentioned in the paper? I even cannot find a reference of it.

W2: Although the author has conducted extensive experiments on many datasets, there are few baselines chosen. The claim regarding the performance of the proposed method in the paper is not convincing enough.

W3: The experimental results in Figure 3 show a significant improvement in time efficiency of the proposed method after reformulating Equation (10). However, kernel methods are often computationally intensive, which restricts their application. The computational complexity analysis between the proposed method and RWK is missing.

Suggestions for improvements:
1, The appendix provides valuable information regarding the experimental settings. However, it does not clearly specify the settings for the GCNConv method. Table 8 indicates that the GCNConv method demonstrates exceptional performance in node classification. Could you please provide additional details regarding the GCNConv method itself, as well as the corresponding experimental settings?

**Ethics Review Description:**

N.A.

**Reviewer Confidence:**

4: The reviewer is certain that the evaluation is correct and very familiar with the relevant literature

**Scope:**

4: The work is relevant to the Web and to the track, and is of broad interest to the community

---

### Official Review · Reviewer_n9qD · 2023-11-27

**Novelty:** 5
**Technical Quality:** 3

**Review:**

Strengths:

1. The paper proposed a learnable random walk kernel, which is more effective than GCN in a few tasks.
2. The paper is well organized and the application of the proposed kernel is well illustrated and tested.

Weaknesses:
1. The baselines are too weak. There are stronger kernels such as WL subtree kernels and Wasserstein WL kernels, which should be compared. This is actually my biggest concern. There is no evidence that we have to use the proposed kernel RWK. We have better choices.
2. The learnable parameters of RWK haven't been clearly explained.

**Questions:**

Please refer to the review.

**Reviewer Confidence:**

4: The reviewer is certain that the evaluation is correct and very familiar with the relevant literature

**Scope:**

3: The work is somewhat relevant to the Web and to the track, and is of narrow interest to a sub-community

---

### Decision · Program_Chairs · 2024-01-22

**Decision:**

Accept

**Comment:**

Although the reviews for this paper were mixed, in my opinion this is a solid work. The strength of the proposed approach is an interesting technique involving color-matching random walks for learnable RW graph kernels, which mitigates issues associated with prior methods that only work under certain restricted conditions and have inefficient parametrization. The authors clarified some concerns about comparisons with other learnable kernel approaches (including WL subtree and Wasserstein WL kernels) which are not differentiable and are currently not scalable. The experimental results are, in my opinion, convincing enough.

 Please make sure to include into your revision the clarifications for simulation experiments requested by some of the reviewers, as well as the additional results based on their valuable comments.